# Towards an Optimal Footprint Based Area Coverage Strategy for a False-Ceiling Inspection Robot

**DOI:** 10.3390/s21155168

**Published:** 2021-07-30

**Authors:** Thejus Pathmakumar, Vinu Sivanantham, Saurav Ghante Anantha Padmanabha, Mohan Rajesh Elara, Thein Than Tun

**Affiliations:** 1Engineering Product Development Pillar, Singapore University of Technology and Design (SUTD), Singapore 487372, Singapore; pathmakumar_thejus@mymail.sutd.edu.sg (T.P.); vinu_sivanantham@sutd.edu.sg (V.S.); saurav@sutd.edu.sg (S.G.A.P.); 2Oceania Robotics Private Limited, 3 Soon Lee Street, # 01-03 Pioneer Junction, Singapore 627606, Singapore; tun_theinthan@oceaniarobotics.com

**Keywords:** functional footprint, false-ceiling inspection, multi-objective optimization, pest-control robot, path-planning

## Abstract

False-ceiling inspection is a critical factor in pest-control management within a built infrastructure. Conventionally, the false-ceiling inspection is done manually, which is time-consuming and unsafe. A lightweight robot is considered a good solution for automated false-ceiling inspection. However, due to the constraints imposed by less load carrying capacity and brittleness of false ceilings, the inspection robots cannot rely upon heavy batteries, sensors, and computation payloads for enhancing task performance. Hence, the strategy for inspection has to ensure efficiency and best performance. This work presents an optimal functional footprint approach for the robot to maximize the efficiency of an inspection task. With a conventional footprint approach in path planning, complete coverage inspection may become inefficient. In this work, the camera installation parameters are considered as the footprint defining parameters for the false ceiling inspection. An evolutionary algorithm-based multi-objective optimization framework is utilized to derive the optimal robot footprint by minimizing the area missed and path-length taken for the inspection task. The effectiveness of the proposed approach is analyzed using numerical simulations. The results are validated on an in-house developed false-ceiling inspection robot—Raptor—by experiment trials on a false-ceiling test-bed.

## 1. Introduction

Pest management is one of the critical elements in facility management that has shown growth to USD 22.7 billion in 2021 and is estimated to reach USD 29.1 billion by 2026 globally [1,2]. The Rodent Monitoring System (RMS) is one of the necessary tasks in a facility management system since the rodents are a critical transmission vector for various diseases [3,4,5,6]. In addition, the infestation of rodents brings loss of revenue from damaged appliances, structural damages, loss of commodities, and expenses for involving pest-control services and professionals [7,8,9].

There have been a variety of rodent inspection techniques developed over time to check for any rodent infestation. An Internet of Things (IoT)-based rodent infestation monitoring system is reported in [10], where the authors utilized sensor-based and vision-based algorithms for rodent monitoring. Ross et al. reported the development of a smart rodent monitoring system called RatSpy that uses computer vision and IoT for inspection within a rodent bait station [11,12]. Sowmika et al. discuss a method of rodent detection in farmlands, where a PIR sensor is used to detect the motion of rodents within a range of ten meters, and sensor data collected are pushed to the cloud for enabling remote monitoring [13]. Similarly, the work mentioned in [14] uses wireless sensor networking for rodent detection. The research outcome disclosed in [15] describes the development of rodent bait equipped with electronic sensors in its bed to detect the presence of rodents within the bait. The prior-art study shows that sensing techniques, IoT, and computer vision are catalysts for the proliferation of rodent inspection techniques. However, the advancements in IoT-based techniques for rodent management and infestation control cannot be considered a perfect solution. The omnipresent rodent species, such as Norway rats (Rattusnorvegicus) and black rats (Rattusrattus), and their strong and consistent behavioral characteristics, such as neophobia [16,17], increase their awareness of trap locations, and so they keep changing their harborage. Furthermore, it is extremely challenging to locate a dead and decaying rodent inside a false ceiling after rodenticide consumption. The scenario mentioned above gives rise to the need to formulate effective methods for rodent activity monitoring inside a false ceiling. In addition to the rodent inspection, another important task to be carried out is the inspection of building structures like optic-fiber cables, sprinkler and gas pipes, and air-condition vents prone to rodent gnaw bites as part of the periodic maintenance. The conventional method for the inspection task is a manual inspection done by humans. This method is ineffective since the person who inspects the false ceiling gets an eclipsed vision by obstacles most of the time. Hence, the manual inspection methods are inefficient and ineffective in terms of time consumption and inspection quality. Furthermore, the risk factor involved in climbing up to high-rise ceilings multiple times escalates the drawbacks of manual false ceiling inspection.

A vast volume of research works is reported on the development of strategies for automated inspection tasks for complex environments [18,19,20,21]. Robot-aided inspection is one of the key factors among automated inspection, which has been widely researched recently. For instance, work mentioned in [22] reports the development effort of flexible pipe inspection robots, where authors propose a bio-inspired caterpillar-like robot for piping inspection. Similarly, Imran et al. developed an inspection robot capable of passing through complicated pipelines with full control [23]. Similar to pipeline inspection, autonomous robots are being used for handling high-risk inspection tasks inside nuclear reactors, where authors propose a lightweight robot with a holonomic base and a manipulator [24]. The development of a novel drainage inspection robot is mentioned in [25], where authors propose strategies for the collision avoidance and stability control of the drainage inspection robot. Additionally, an inspection of contacts on overhead cable systems for high-speed rails using inspection robots is discussed in [26]. Leveraging on the precedence on effective usage of robots for complex inspection strategies and the need to eliminate risk and inefficiency factors involved in false-ceiling inspection gives rise to a new modality of inspection, which is the usage of lightweight inspection robots. A robot that is capable of navigating around the false ceiling either in an autonomous or semi-autonomous fashion can perform rodent activity monitoring and periodic false ceiling inspection efficiently. However, the brittle nature of false-ceiling panels installed inside a building restricts heavy structural weight. To maximize accessibility, the dimensions of the robot have to be compact. The design considerations mentioned above on structural weight and dimensions do not allow the robot to carry a heavy battery on board. Hence, the inspection coverage planning has to be efficient in terms of energy consumption.

Autonomous navigation is an essential factor for autonomous robots. There is numerous research work reported on perception, navigation, and localization methods [27,28,29,30,31]. Among all, a significant research focus is given on the path planning strategy since it can have an impact on the task performed by the robot. For instance, the work mentioned in [32] describes the path-planning approach in a mobile robot for a gas mapping task. Miao et al. proposed a scalable coverage planning strategy for a cleaning robot [33]. From the author’s perspective, a path-planning approach for area coverage is a critical element in robotic cleaning. Sasaki et al. proposed an adaptive path planning strategy for improving the performance of a cleaning robot, where the coverage planning prioritizes and sorts regions based on dirt distribution [34]. The footprint of the robot is a crucial element of area coverage algorithms. A classic robot footprint is defined as the surface area occupied by the robot at any instant of time [35]. For complete coverage planning, the robot should be navigating so that the defined robot footprint will cover the maximum possible area. The same strategy is applied for complete visual coverage tasks as well [36]. Hence, for a false ceiling inspection, the effectiveness of the task depends on the robot’s ability to cover every location in a false ceiling. For a visual inspection, the robot footprint can be defined as the maximum area within the camera’s field of view that can be inspected. However, the efficiency of inspection also has an impact on the choice of an optimal footprint.

The robot footprint used to maximize the task efficiency is also defined as the functional footprint of the robot [37]. In this paper, we define an optimal footprint, also known as functional footprint, for an in-house developed false ceiling inspection robot called Raptor. In the scenario mentioned above, the robot’s functional footprint depends upon the position and orientation of the camera onboard. We have exploited genetic algorithms to determine the functional footprint of the robot that ensures an efficient false-ceiling inspection. The rest of the research article is organized as follows:

Section 2 details the general objective of this work, followed by the system architecture of the Raptor robot in Section 3. Section 4 explains the coverage planning strategies used by the Raptor robot, and Section 5 gives a detailed insight regarding the cost functions, constraints, and optimization strategies for finding the functional footprint. The results of the simulation are given in Section 6, and finally, the conclusion and future works are given in Section 7.

## 2. Objectives

The functional footprint of the robot is the footprint area of the robot that enhances the efficiency of task execution [37]. The general objective of this work is to define a functional footprint for the Raptor robot and improve the performance of the inspection task the robot carries out by considering the defined functional footprint. This general objective is subdivided into three components:A zig-zag area coverage strategy is proposed for false-ceiling inspection, where the area coverage is defined in terms of the robot footprint, which is dependent on the position and orientation of the camera (camera installation parameters).Formulate cost functions based on area missed and path-length covered during the inspection by the robot.Estimate the functional footprint by finding the optimal values for camera installation parameters by minimizing the costs using multi-objective optimization.Validate the results of the Raptor robot by conducting experiment trials on a false ceiling test-bed.

## 3. System Architecture

The outline of the in-house developed Raptor robot and its associated components are given in Figure 1. The robot was designed by considering the maximum supported weight on a false ceiling of 2.0 kg. The overall weight of the robot with a 2800 mAH LiPo battery is 1.2 kg, and the weight without a battery is approximately 1.03 kg. The developed robot system comprises four main segments: Operator’s Console Unit, ROS master controller, Payload platform, and Beacon platform. The ROS master control unit controls the movement of all deployed robots at any given time. There are two variants for the Raptor robot platform, namely: Payload platform and Beacon platform. The payload platform can perform semi-autonomous navigation using the RPlidar A1M8 LIDAR sensor as the primary sensor for localization and navigation. On the other hand, the beacon platform works with UWB-based mobile and stationary beacons for indoor localization. The Operator’s Console Unit is the Graphic User Interface (GUI) layer to interface with the ROS master. The ROS master synchronizes the platform data between the beacon platforms and the payload platforms. The platform has the dimensions: 390×350×200 mm. We will focus on the Beacon platform for the false ceiling inspection task since we require map-less and GPS-denied indoor navigation. The beacon platform has a skid-steering base powered by four DC Motors that operate on a closed-loop velocity control. A pair of Roboclaw motor controllers drive the motor according to the high-level command control received from a single board computer. Raspberry Pi-4 Model B is used as the slave single-board computer for processing of high-level computation, communication, and control with the remote ROS master controller. The motor driver and Raspberry Pi controller are placed inside a unique chassis to avoid interference induced by the payload and DC motors. One Roboclaw controls the front two motored wheels while another drives the back two motored wheels. Each Roboclaw is connected to a pair of motors with encoders. A DC–DC buck-converter that steps down to 5 V is used to power a Raspberry Controller. The Raspberry Pi is connected to the two Roboclaws. The data from the Inertial Measurement Module(IMU) and wheel-odometry information from Roboclaw are used for dead-reckoning. Global localization of the robot is handled by the Marvel-minds beacon and GPS module associated with the robot. A Marvelmind indoor navigation beacon that is advantageously placed at the top center of the robot provides the necessary position information for GPS-denied navigation. A spring-return-based wheel suspension mechanism, designed according to the dimensions and kinematics of the robot, allows the robot to overcome high-rise obstacles and improves overall platform stability while navigating.

### Control and Autonomy

The overall control architecture of the Raptor system is shown in Figure 2. The software nodes for the control and autonomy tasks are implemented over a Robot Operating System (ROS), running across a ROS master and integrates stationary beacons and other robot platforms. The ROS master controller is the primary controller that controls and overviews the rest of the sub-modules in the robot platform. A central workstation hosts the ROS master, and the beacon platforms run as slave devices. The slave devices are the ROS-enabled Raspberry Pi nodes that are in each robot platform. A user can establish a connection with the ROS master using a mobile or computer application over WiFi. The UWB (Ultra Wide Band) outdoor beacons by Marvel mind, an indoor positioning system, was used to find the robot location accurately up to 2 cm resolution. The positioning system works a minimum of 3 stationary beacons placed at different corners of the test-bed and a mobile beacon placed on top of the robot platform. Each beacon can either transmit or receive a signal. Therefore, Marvelmind has developed two localization architectures: Inverse Architecture (IA) and Non-Inverse Architecture (NIA). In IA, stationary beacons act as transmitters, and mobile beacons act as receivers with a maximum localization update rate of 16 Hz, halved with each additional mobile beacon added. In NIA, stationary beacons act as receivers, and mobile beacons act as transmitters, with the localization update rate remaining the same, albeit with a more complicated setup. This platform uses the IA approach for autonomous navigation. In addition to the UWB beacon, IMU and wheel odometry data are also used for the robot localization.

## 4. Path-Planning Strategy

The task of the Raptor robot is to inspect the maximum area in a false ceiling using a camera for rodent activities. Hence, the robot follows a vision-based complete coverage planning strategy (CCP). There are multiple approaches for achieving the CCP [38,39,40]. However, this study exploits the classic zig-zag path planning strategy to achieve maximum coverage for an area devoid of obstacles.

### 4.1. Robot Footprint Definition

Conventionally, the mobile robot’s footprint is the area swept by the robot when it is stationary. For an inspection robot, the footprint is defined as the effective area that can be inspected while stationary. In other terms, the footprint is the camera’s effective field of view. By definition, [37], the functional footprint is the optimal footprint associated with the robot that maximizes the task efficiency. Hence, the functional footprint ($f_{r}$) can be considered as the optimal inspectable area using the usable field of view of the camera. On the other hand, the performance and efficiency of an inspection task can be weighed using the time taken for inspection, the area missed during the inspection, and the energy consumed by the robot during the inspection. Under the notion that traveling a minimal path for a given area will yield minimum time and minimal energy consumption, we considered area missed and path-length as the efficiency-determining factors in the scenario. To obtain the functional footprint of the robot, dependency on the functional footprint on every efficiency determining factor has been investigated. The parameters that define robot footprint are:Position of camera mounting in both the x and y-axis.The horizontal mounting angle α (yaw angle).The vertical mounting angle β (pitch angle).

Therefore, the $f_{r}$ (functional footprint) is represented by the optimal values for the camera’s position ($x^{*}$ and $y^{*}$) and orientation (yaw angle—$\alpha^{*}$ and pitch angle—$\beta^{*}$) that results in the minimal area missed minimal total path length to be traveled during the inspection. Hence, the functional footprint can be represented as in Expression (Equation 1) given below:(1)$$f_{r}=\left\{\begin{array}{c} x^{*}\\ y^{*}\\ \alpha^{*}\\ \beta^{*} \end{array}\right\}$$

The superscript ∗ on the variables represents that they hold optimal values.

### 4.2. Zig-Zag Based Area Coverage

Given a rectangular cross-section of size L×B, the whole region is divided into grid cells of size l×b. The size of the grid cells specified is split with respect to the camera’s field of view; the robot moves along each row, covering all the grid cells with the defined functional footprint of the camera. At the end of each row, the robot turns 90 degrees and moves to the next grid cell to reach the next row. This is repeated until the robot covers all the grid cells specified in the region up to the last row.

## 5. Optimal Coverage Strategy

This section details the derivation of the necessary objective functions for the minimum distance traveled while maximizing the area coverage for the false ceiling inspection using a zig-zag-based coverage strategy.

### 5.1. Total Zig-Zag Path Length Computation

Figure 3 shows the side-view of the robot from which we can geometrically extract the grid size for zig-zag path in terms of θ, β, and *z*. In Figure 3, *z* is the height at which the camera is positioned on the robot from the floor, β is the vertical angle of the camera, and θ represents the horizontal field of view of the camera. The length of the grid *l* can be calculated in terms of β and θ using Equation (Equation 2):(2)$$l=z\;tan(\beta +\frac{\theta }{2})\;;$$

Figure 4 shows the top view of the robot with $X_{r}$ and $Y_{r}$ as the x and y axes with respect to the robot frame. From Figure 4b, the breadth of the grid *b* can be geometrically derived. The *b* can be calculated in terms *l* and θ using Equation (Equation 3):(3)$$b=2l\times tan(\frac{\theta }{2})$$

For the zig-zag coverage strategy, the path length can be calculated in terms of distance covered in each row and distance covered during the transition between the rows, as shown in Figure 4a. Hence, the total path length of the robot can be represented using Equation (Equation 4):(4)$$S_{total}=\left(S_{rows}\times n_{rows}\right)+\left(S_{transition}\times \left(n_{rows}-1\right)\right)$$
where, $S_{rows}$ is the total distance covered along the rows, $S_{transition}$ is the distance covered during the transition between the rows and is given by Equations (Equation 5) and (Equation 6):(5)$$S_{rows}=\left(Length_{nrow}-\left(\frac{b}{2}\;sin\left(\alpha \right)+l\;cos\left(\alpha \right)+y\right)\right)n_{rows}$$
where *y* is the position of the camera in the y-axis and α, β are the orientation of the camera on the robot, $Length_{nrow}$ is the total length of each row strip, and $n_{rows}$ is the total number of row strips in the grid space. The term $\frac{b}{2}\;sin\left(\alpha \right)+l\;cos\left(\alpha \right)$ is the total horizontal distance covered by the robot with the change in orientation angle α of the camera.
(6)$$S_{transition}=\left(2\;b\;tan\left(\theta /2\right)\right)\left(n_{rows}-1\right)$$

$n_{rows}$ is the total number of rows in the area of operation. The number of rows $n_{rows}$ can be calculated using Equation (Equation 7):(7)$$n_{rows}=\frac{B}{b}$$

The total distance ($S_{total}$) covered by the robot for maximum area coverage can be written in terms of *y*, α and β using Equation (Equation 8): (8)$$S_{total}\left(y,\alpha ,\beta \right)=\left(Length_{nrow}-\left(\frac{b}{2}\;sin\left(\alpha \right)+l\;cos\left(\alpha \right)+y\right)\right)n_{rows}+\left(2\;b\;tan\left(\theta /2\right)\right)\left(n_{rows}-1\right)$$

From Equation (Equation 8), it is clear that the path length of the robot is dependent on parameters that define the footprint of the robot, which is α, β, and *y*. Hence, the functional footprint of the robot can be determined by estimating the optimal footprint parameters for the minimal path length for a given area of operation.

### 5.2. Total Area Missed

During the zig-zag path planning, loss in area coverage happens in two cases:The robot moves along the horizontal path while covering a row (Figure 4b).The robot does a transition between rows (Figure 4c).

The total area missed is formulated as the sum of the areas missed in case 1 and case 2. For case 1, the area missed is the initial area loss due to the robot’s starting position combined with area loss accumulated while the robot moves in the horizontal path. The initial area loss at the starting position can be estimated by calculating the areas represented by section 1, section 2, section 3, and section 4 represented in Figure 4b shown in Figure 4b. The area of section 1 can be estimated by calculating the area of the trapezoid as per Equation (Equation 9):(9)$$Area_{section1}=\frac{\frac{b}{2}\;cos\left(\alpha \right)+l\;sin\left(\alpha \right)}{2}\times \left(\frac{b}{2}\;sin\left(\alpha \right)+l\;cos\left(\alpha \right)\right)$$
where $\frac{b}{2}\;cos\left(\alpha \right)$ and lsin(α) are the two sides of the trapezoidal region and $\frac{b}{2}\;sin\left(\alpha \right)+l\;cos\left(\alpha \right)$ is the total horizontal length of section 1. Similarly, the area under section 2 can be calculated using the area of the rectangular region given in Equation (Equation 10):(10)$$Area_{section2}=\left(\left(\frac{b}{2}sin\left(\alpha \right)+l\;cos\left(\alpha \right)\right)\times x\right)$$

The triangular area in section 3 is not considered in the estimation of initial area loss as most of the region in section 3 will be covered when the robot moves along the second row strip in a zig-zag path by overlapping a small region of the adjacent row strip. The accumulated area loss during the robot motion can be approximated with the area of section 4. The area of section 4 can be estimated using Equation (Equation 11)
(11)$$Area_{section4}=\left(n_{rows}\times \left(Length_{nrow}\right)\times \left(l\;sin\left(\alpha \right)+x\right)\right)$$
where $n_{rows}$ are the total number of rows in the grid space, and $Length_{nrow}$ is the length of each row strip.

The area missed during the transition between rows can be approximated as the area of triangle ABC shown in Figure 4c. From Figure 4c, LineA meets with CircleO at point *C*, and LineB meets at the same CircleO at point *B*. Point *A* is the meeting point of LineB and LineA. The representation of LineA is given by Equation (Equation 12):(12)$$Y-\frac{3\times b}{2}=0,$$
and the representation of LineB is given by Equation (Equation 13):(13)X−(l+y)=0

The equation for the CircleO can be represented as:(14)$${\left(x-h\right)}^{2}+{\left(y-k\right)}^{2}={\left(l+y\right)}^{2}$$
where, *h* and *k* are the origins of the robots coordinates. The equation of the triangle ABC in terms of point *A*, *B* and *C* can be written as:(15)$$Area_{triangle}=\frac{1}{2}\left(x_{}1\left(y_{2}-y_{3}\right)+x_{2}\left(y_{3}-y_{1}\right)+x_{3}\left(y_{1}-y_{2}\right)\right)$$
where, ($x_{1},y_{1}),\left(x_{2},y_{2}\right)$ and $\left(x_{3},y_{3}\right)$ are the coordinates of points (A, B, C), which are obtained by solving a set of Equations (Equation 12)–(Equation 14). The total area missed, Equation (Equation 16), can be determined from Equations (Equation 9)–(Equation 11) and (Equation 15):(16)$$\begin{array}{c} A\left(x,\alpha ,\beta \right)=\left(\frac{\frac{b}{2}\;cos\left(\alpha \right)+l\;sin\left(\alpha \right)}{2}\right)\left(\frac{b}{2}\;sin\left(\alpha \right)+l\;cos\left(\alpha \right)\right)+(\frac{b}{2}sin\left(\alpha \right)+l\;cos\left(\alpha \right))x+\\ n_{rows}\left(Length_{nrow}\left(l\;sin\left(\alpha \right)+x\right)\right)+\left(\left(n_{rows}-1\right)\left(Area_{triangle}\right)\right) \end{array}$$

Similar to the equation for path-length (Equation 8), we can clearly observe that the area missed by the robot (Equation 16) is dependent on parameters that define the footprint of the robot, which are α, β, and *x*. Hence, the estimation of the functional footprint boils down to a multi-objective optimization problem with objectives given in Equation (Equation 17):(17)$$min\left(A\left(x,\alpha ,\beta \right)\right)\;\;and\;\;min\left(S_{total}\left(y,\alpha ,\beta \right)\right)$$
subject to the constraints:(18)$$x_{min}\leq x\leq x_{max};\;\;y_{min}\leq y\leq y_{max};\;\;\alpha_{min}\leq \alpha \leq \alpha_{max};\;\;\beta_{min}\leq \beta \leq \beta_{max}.$$
where $x_{min}$, $x_{max}$ and $y_{min},y_{max}$ are the minimum and maximum possible position of camera on the robot; $\alpha_{min},\alpha_{max}$ and $\beta_{min},\beta_{max}$ are the minimum and maximum possible yaw angle and pitch angle to adjust the horizontal and vertical field of views. The parameters that shape the zig-zag path: (1) number of rows $n_{rows}$ (2) row length $S_{row}$ and (3) transition length $S_{transition}$ are computed from the optimal values for α, β, *x* and, *y*. The multi-objective optimization problem defined by Equations (Equation 17) and (Equation 18) is solved using a Genetic Algorithm (GA). GA is a subset of evolutionary algorithms that are widely used for solving complex multi-objective optimization problems. However, the genetic algorithms are modified to accommodate multiple objective functions for multi-objective optimization problems. There are different Multi-Objective Evolutionary algorithms (MOEA), including NSGA2 [41,42,43,44] and PESA-II [45,46]. In this work, we focused on NSGA-2 and MOEA/D algorithms for solving the constraint multi-objective optimization problem.

#### 5.2.1. NSGA-2

The procedures involved in the NSGA2 algorithm are explained in the flow chart given in Figure 5. The NSGA2 algorithm yields a set of solution points, also known as the Pareto-Optimal solution set. The Pareto-front holds the dominant solutions for conflicting objectives functions. The NSGA2 algorithm is known for its simplicity of implementation and effectiveness. The NSGA2 algorithm is an improved version of NSGA using the fast non-dominated sorting concept [41].

#### 5.2.2. MOEA/D

The multi-objective evolutionary algorithm based on decomposition (MOEA/D) works by decomposing a given problem to multiple neighborhood sub-problems and finding the optimal solutions simultaneously [47,48]. MOEA/D is known to be more straightforward compared to other multi-objective optimization algorithms. Hence, we considered the usage of MOEA/D alongside NSGA2 for performing the concerned multi-objective optimization problem. The procedures involved in the MOEA/D algorithm are explained in the flow chart given in Figure 6.

## 6. Results and Discussion

For the given scenario, we exploited genetic algorithms (GA) using NSGA2 (Non-Dominant Sorting Genetic Algorithm). We have conducted multiple simulations with different variations in population size of NSGA2 to solve for the functional footprint of the robot. In addition to NSGA2, a multi-objective evolutionary algorithm based on decomposition (MOEA/D) was also used to arrive at an optimal solution. The simulations were performed using the Pymoo library for NSGA2 and MOEA/D in Python3. The algorithms were executed on a computer with an Intel Core-i7 CPU and 16GB RAM. Part of the Section details the multiple simulation trials and experiments that were conducted.

### 6.1. NSGA2—Variation of Population

The NSGA2 optimization was completed for three different population sizes; 250, 500 and 2000. For the MOEA/D, the number neighbors and probability of neighbor mating are fixed as 15 and 0.7, respectively. For the area coverage, different area sizes of 2×2, 3×2 and 4×2 m in a false ceiling environment were considered for experimentation. In each set of simulations, the upper and lower bound of the constraints $\alpha_{max}$, $\alpha_{min}$, $\beta_{max}$, $\beta_{min}$, $x_{max}$, $x_{min}$, $y_{max}$, $y_{min}$ were chosen as (0.0, 0.9547), (0.0, 0.687), (−0.15, 0.15) and (−0.10, 0.0) respectively. The simulation results for population 250 is given in Figure 7. The number of points belonging to the Pareto-front is less for a population of 250 compared to the other set of simulation trials. In the design space, the solutions were widely spread between (−0.145, 0.142), (−0.0941, −0.014), (0.0013, 0.499) and (0.019, 0.685) for *x*, *y*, α, and β. The simulation results for a population of 500 are given in Figure 8. The number of points corresponding to the Pareto-front is more for a population of 500 compared to a population of 250 in the previous simulation. In the design space, the solutions were spread across the intervals of (−0.1498, −0.00485), (−0.0939, −0.0067), (0.0002, 0.117) and (0.136, 0.6859) for *x*, *y*, α, and β. However, compared to simulation for a population of 250, the spread of solutions mostly converged to values −0.1498, −0.0939, 0.00 and 0.685 for *x*, *y*, α, and β. When it comes to the population of 2000, (given in Figure 9), the number of points belonging to the Pareto-front is more than that of simulations corresponding to populations of 250 and 500. In the design space, the solutions were less spread between (−0.1499, 0.0866), (−0.099, −0.0037), (0.00004, 0.365) and (0.124, 0.685) for *x*, *y*, α, and β. The values for *x*, *y*, α, and β are more converged to −0.1499, −0.0999, 0.003 and 0.685.

### 6.2. MOEA/D

The simulation results for the MOEA/D algorithm for a neighboring size of 15 are given in Figure 10. In the design space, the solutions were widely spread between (−0.1498, 0.0149), (−0.0931, −0.00), (0.0063, 0.499) and (0.0575, 0.685) for *x*, *y*, α, and β. Figure 11 shows the overall time taken to run each of the simulations by varying the population size and changing the optimization algorithm.

Table 1 summarizes the simulation results for varying population sizes in the optimization algorithm. The simulation for NSGA2-2000 takes at least 70% more time than the other optimization algorithm types. However, an optimization with a population size of 2000 for NSGA2 has shown better convergence results when compared with other population sizes and the MOEA/D algorithm. Therefore, we considered using NSGA2-2000 for further simulations by varying the area of the test-bed.

### 6.3. NSGA-2 Variation of Area

The NSGA2-2000 optimization was completed for three different test-bed sizes of 2 × 2, 3 × 2 and 4 × 2 m. For each of the simulation results, total area missed and total path length for the robot to maximize area coverage were computed. Figure 12, Figure 13 and Figure 14 show the Pareto-chart using NSGA2-2000 for each of the area sizes. The solutions for all of the test-bed sizes were less spread between (−0.1499, 0.006), (−0.0999, −0.0005), (0.0002, 0.378) and (0.023, 0.6859) for *x*, *y*, α, and β. The values for *x*, *y*, α, and β are more converged to −0.1499, −0.0999, 0.0002 and 0.685.

Table 2 summarizes the simulation results with overall area coverage performance and path length of the robot to maximize area coverage using NSGA2-2000 for different test-bed sizes.

### 6.4. Experiments

The experiments were carried out on the Raptor robot to validate the simulation results for the optimal placement of the camera to maximize area coverage and minimize path length. The robot was operated inside a false ceiling test-bed of a total area of 4×2 m. Multiple experiments were performed based by reducing the area of operation to 2×2 (Figure 15), 3×2 (Figure 16) and 4×2 m (Figure 17) within the same test-bed. We installed MarvelMind beacons on different locations of the test-bed for robot localization. The total path length for each trial is recorded from the experiment results. We estimated the robot pose by fusing the beacon position, wheel odometry, and inertial measurements from the IMU sensor. The row length, transition length between the rows obtained after performing the optimization are programmed on the robot for performing the zig-zag.

Table 3 summarizes the experimental data obtained. Time taken for the robot to cover 2×2, 3×2, and 4×2 m are 61.132, 80.177, and 120.43 s, respectively. Correspondingly, path-lengths for 2×2, 3×2, and 4×2 m were 8.042, 12.542, and 15.662 m, respectively. The observed results agree with the simulation results for 2, 3×2, and 4×2 m.

## 7. Conclusions and Future Works

This work calculates an optimal footprint for an in-house developed inspection robot called Raptor, where the footprint is defined by the optimal position and orientation of a camera on the robot. We considered the minimization of area-missed during the inspection task and path length of the robot for different positions and orientations of the camera. The mathematical expression for area missed and path covered for a zig-zag area-coverage strategy was derived, and optimal values for camera position coordinates and orientation angles were determined using constraint multi-objective optimization. We utilized multi-objective optimization tools, such NSGA-2 and MOEA/D, for performing multi-objective optimization. The results of multi-objective optimization were validated by conducting experiments on a false-ceiling test-bed using the Raptor robot. The future works of this research will focus on the dynamic optimization, consideration of energy and multiple path-planning strategies for defining functional footprint-based path generation considering static and dynamic obstacles. 

## Figures and Tables

**Figure 1 sensors-21-05168-f001:**
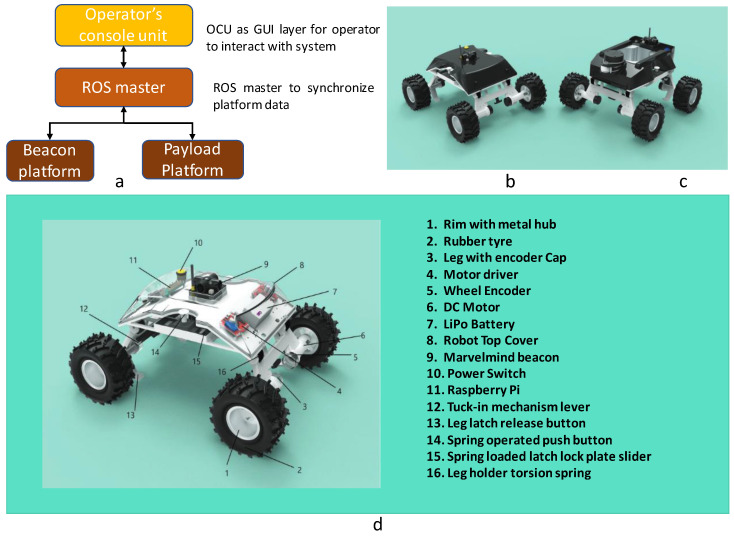
The outline of the Raptor system of robots (**a**), the beacon platform (**b**), the payload platform (**c**), components associated with a beacon platform (**d**).

**Figure 2 sensors-21-05168-f002:**
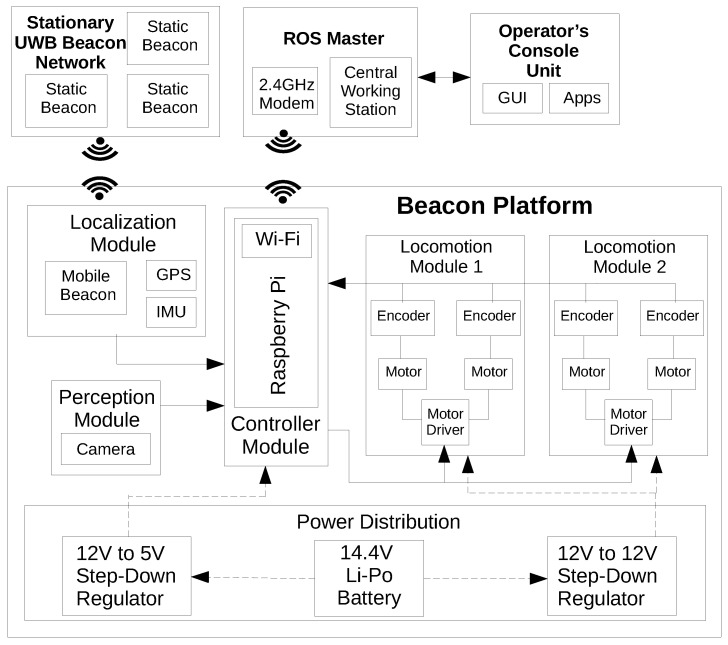
The control architecture of the Raptor robot.

**Figure 3 sensors-21-05168-f003:**
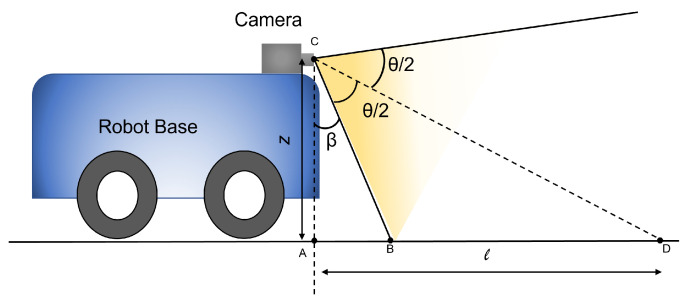
The side-view of the robot with the field-of-view of the camera.

**Figure 4 sensors-21-05168-f004:**
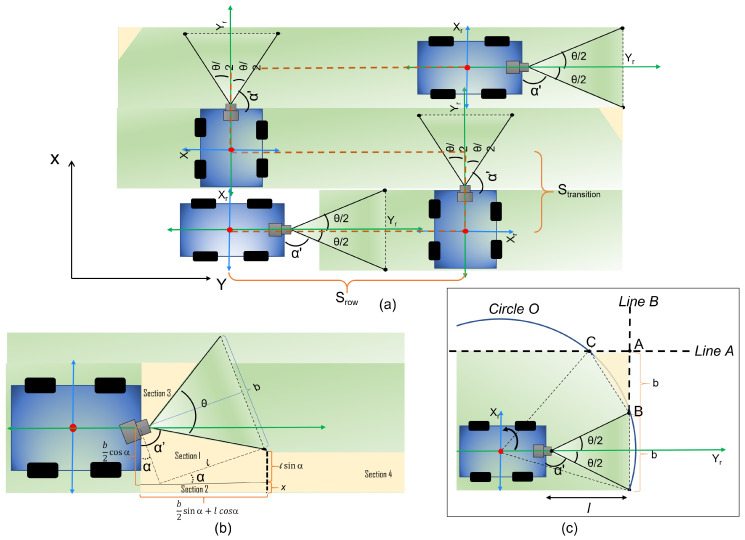
The zig-zag area coverage strategy of the robot (**a**), area missed during the initial stage of operation (**b**), area missed at the turning (**c**).

**Figure 5 sensors-21-05168-f005:**
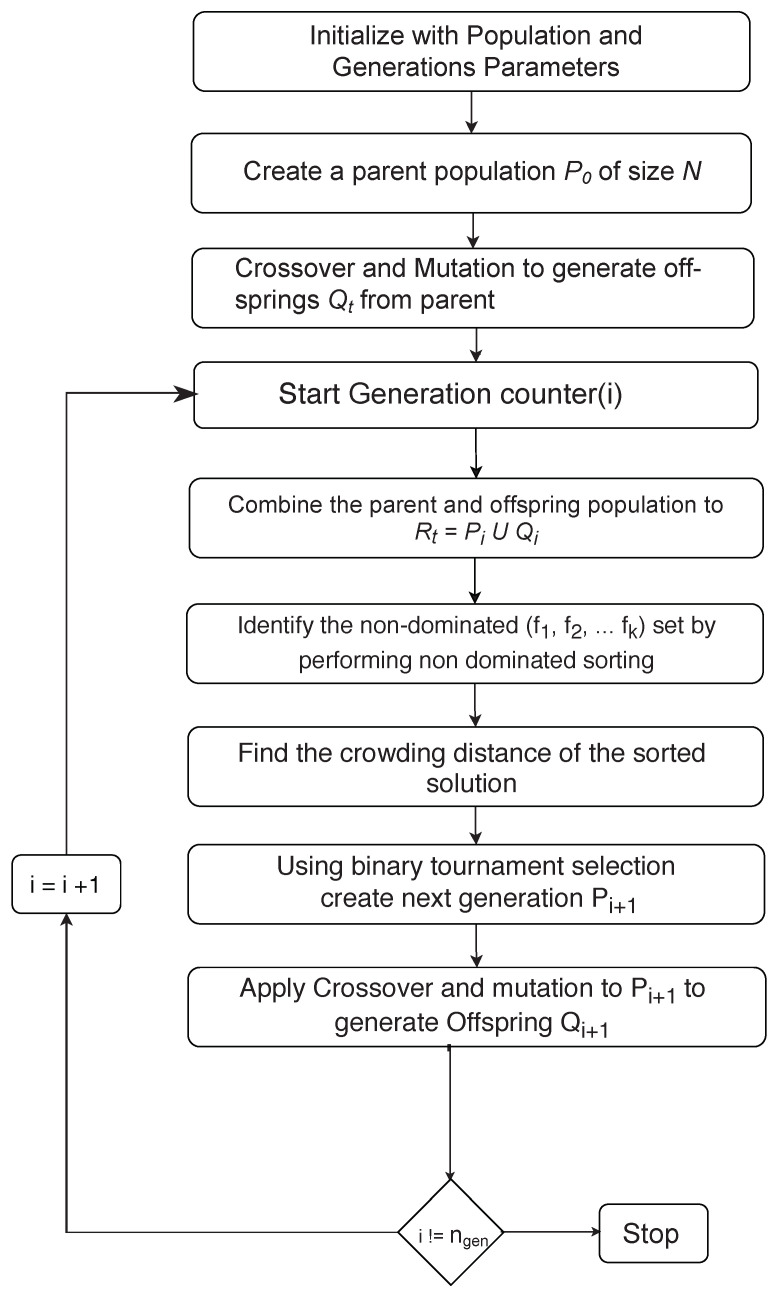
Procedure for the NSGA-2 algorithm implementation [41].

**Figure 6 sensors-21-05168-f006:**
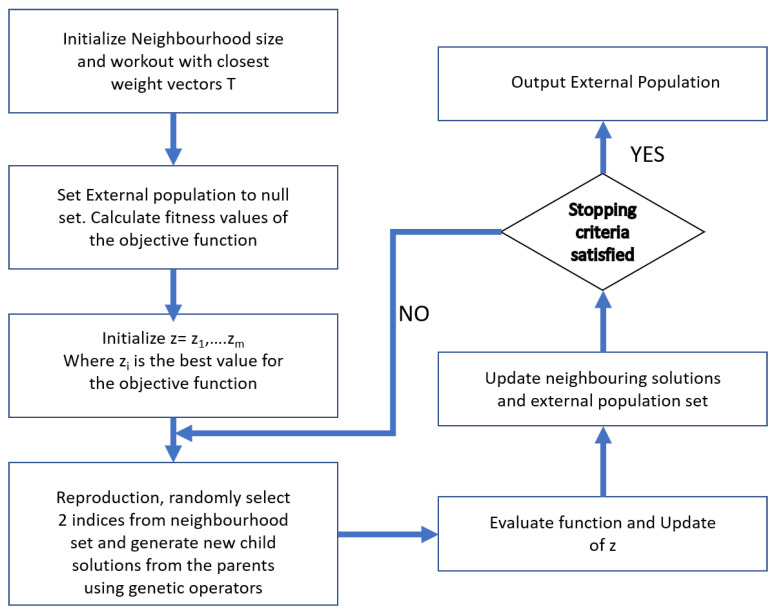
Procedure for MOEA/D algorithm implementation.

**Figure 7 sensors-21-05168-f007:**
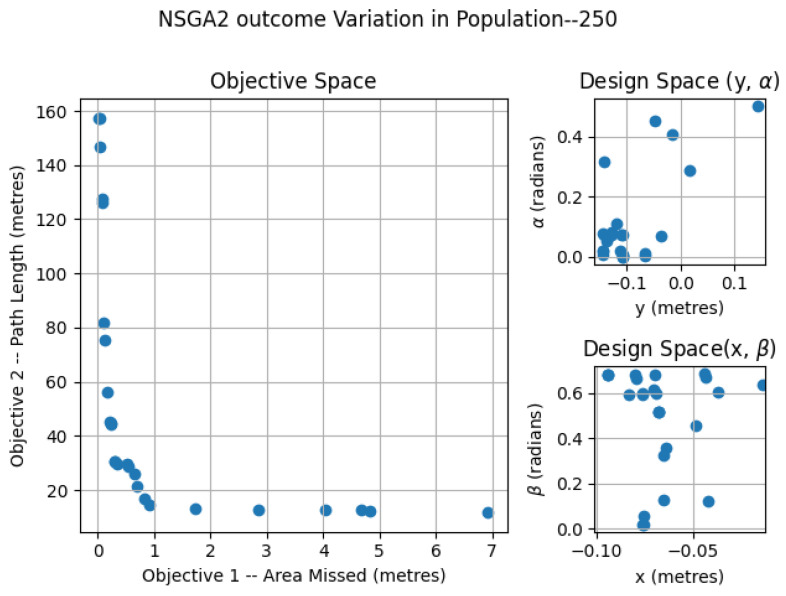
NSGA2 optimization for area size 4 × 2 m with population size = 250.

**Figure 8 sensors-21-05168-f008:**
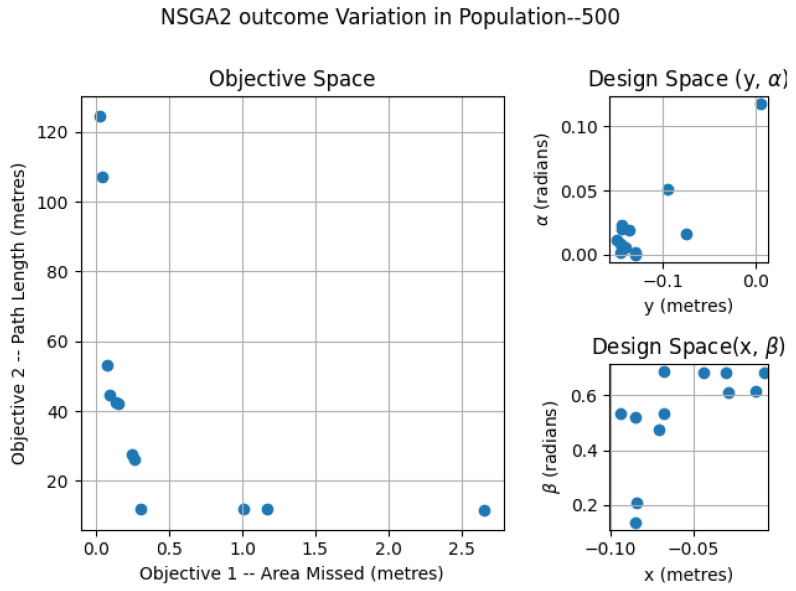
NSGA2 optimization for area size 4 × 2 m with population size = 500.

**Figure 9 sensors-21-05168-f009:**
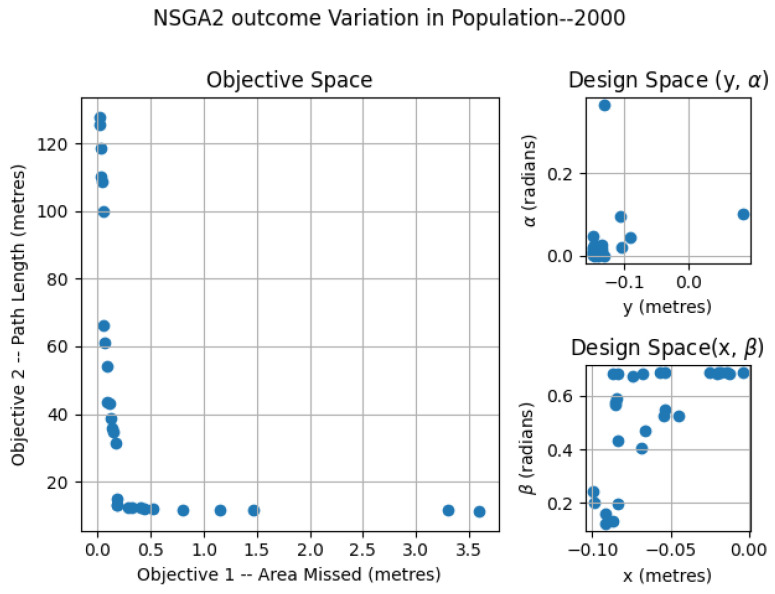
NSGA2 optimization for area size 4 × 2 m with population size = 2000.

**Figure 10 sensors-21-05168-f010:**
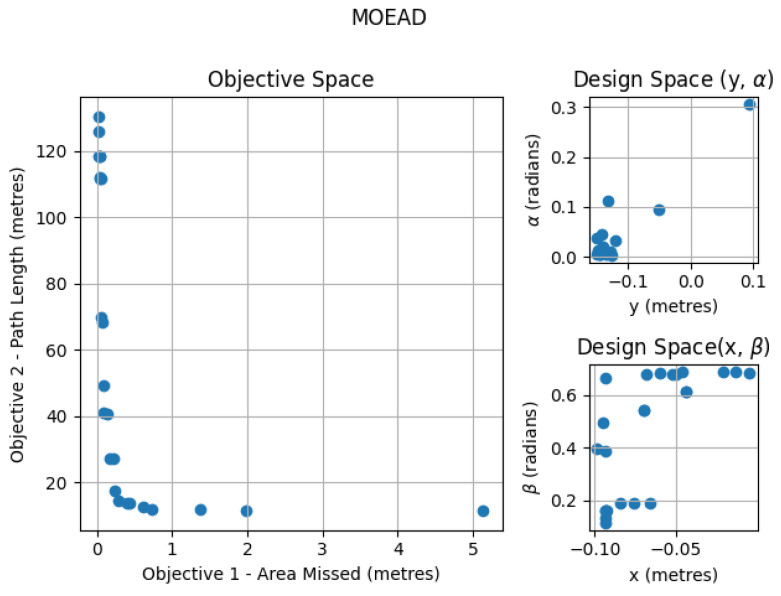
MOEA/D optimization for area size 4 × 2 m with neighboring size 15.

**Figure 11 sensors-21-05168-f011:**
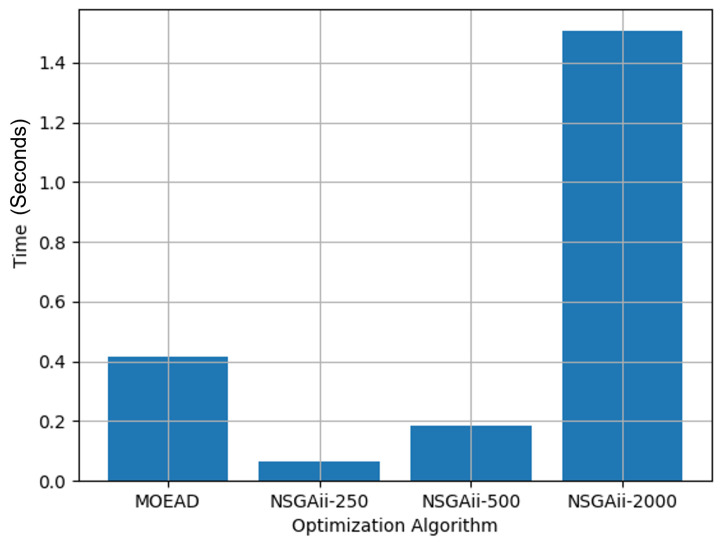
Time comparison for MOEA/D and variations of NSGA2 for finding the Pareto-front.

**Figure 12 sensors-21-05168-f012:**
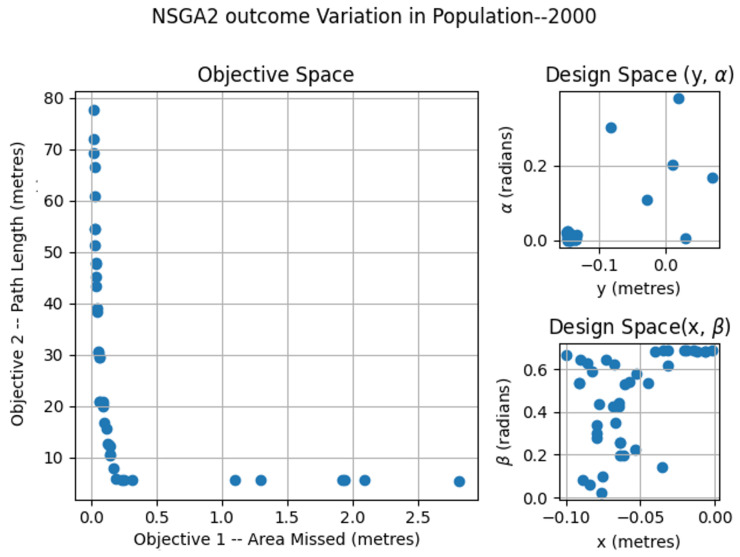
NSGA2 optimization for area size 2 × 2 m with population size = 2000.

**Figure 13 sensors-21-05168-f013:**
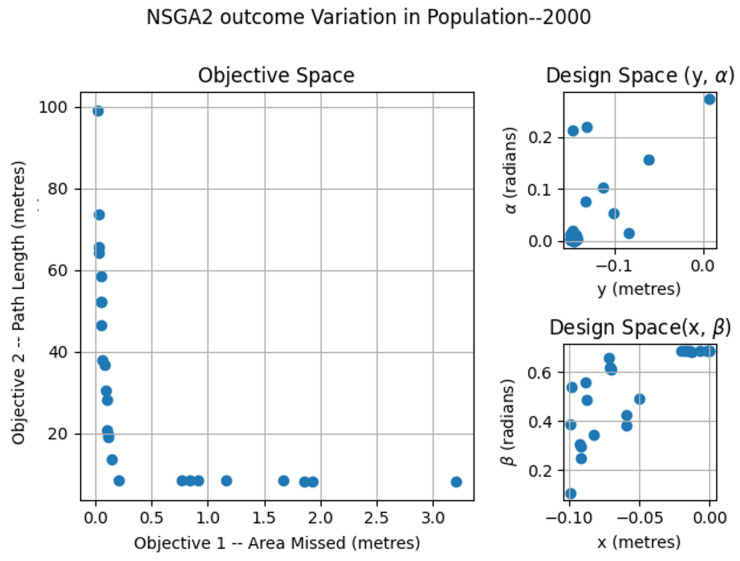
NSGA2 optimization for area size 3 × 2 m with population size = 2000.

**Figure 14 sensors-21-05168-f014:**
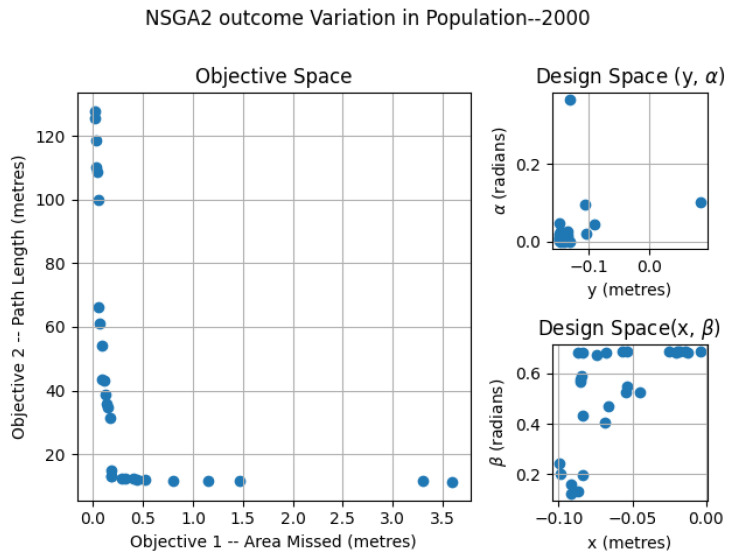
NSGA2 optimization for area size 4 × 2 m with population size = 2000.

**Figure 15 sensors-21-05168-f015:**
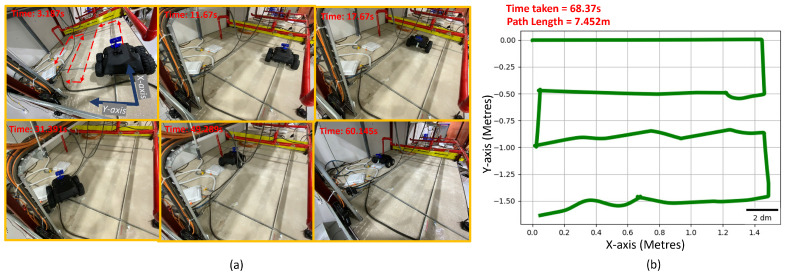
The experiment trial performed in the false-ceiling test-bed for covering 2×2 m. (**a**) Robot performing inspection in the false-ceiling test-bed at various time frames. (**b**) Fused odometry plot of robot during experimental trial.

**Figure 16 sensors-21-05168-f016:**
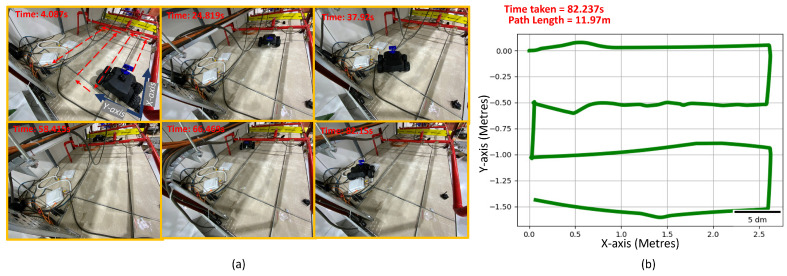
The experiment trial performed in the false-ceiling test-bed for covering 3×2 m. (**a**) Robot performing inspection in the false-ceiling test-bed at various time frames. (**b**) Fused odometry plot of robot during experimental trial.

**Figure 17 sensors-21-05168-f017:**
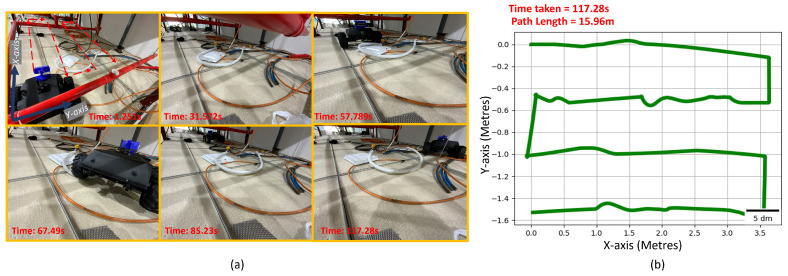
The experiment trial performed in the false-ceiling test-bed for covering 4×2 m. (**a**) Robot performing inspection in the false-ceiling test-bed at various time frames. (**b**) Fused odometry plot of the robot during the experimental trial.

**Table 1 sensors-21-05168-t001:** Consolidated simulation results for varying population size.

Simulation Results for Varying Population Size			
Optimization Algorithm	Time Taken (Seconds)	Area Missed (Metres)	Path Length (Metres)
NSGAii (population size—250)	0.0666	0.87	16.9
NSGAii (population size—500)	0.182	0.332	15.2
NSGAii (population size—2000)	1.424	0.16	14.6
MOEA/D (neighboring size—15)	0.414	0.37	15.7

**Table 2 sensors-21-05168-t002:** Consolidated simulation results for varying area.

Simulation Results for Varying Area Using NSGA2-2000				
Test-Bed Size (Metres)	$S_{\mathrm{row}}$ (Metres)	$S_{\mathrm{transition}}$ (Metres)	Area Missed (Metres)	Path Length (Metres)
2 × 2	1.401	0.4951	0.19	7.089
3 × 2	2.401	0.4951	0.21	11.08
4 × 2	3.401	0.4951	0.16	15.08

**Table 3 sensors-21-05168-t003:** Consolidated experimental results for varying areas.

Experimental Results for Varying Area Using Solutions from NSGA2-2000		
Test-Bed Size (Metres)	Time Taken (Seconds)	Path Length (Metres)
2 × 2	68.37	7.452
3 × 2	82.237	11.97
4 × 2	117.28	15.96

## Data Availability

Data sharing not applicable.

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
