# Peer review of "Towards an Optimal Footprint Based Area Coverage Strategy for a False-Ceiling Inspection Robot"

_sensors, 2021, doi:10.3390/s21155168_

Round 1
Reviewer 1 Report
This paper describes a method to define the optimal camera installation parameters (yaw, pitch and x and y position in the body reference frame) for inspection, with the aim of minimizing the total travelled path and the uncovered area. Along the paper instead of talking about optimal footprint, I would refer to camera installation parameters, which are the resulting outputs of the proposed algorithm.
The paper must be improved by reorganizing the sections and adding some more details to better clarify the proposed algorithm. In addition, there are some conceptual errors that are highlighted in the following comments.
As far as paper organization is concerned, I would introduce the methodology before detailing the setup and system architecture. In addition, the NSGA algorithm should be included in the methodology part, leaving the results and discussion section only describing the result obtained with the proposed methodology.
Please, find in the following some major comments:
- Please define x and y as position of camera in line 193. And α and β in lines 194 and 195.
- A graphical representation of b, defined in equation 1 is needed. Indeed, it is not clear what b is, and how it is estimated as a function of z and θ. In, addition a graphical representation of b, would help the paper comprehension of the other formulas.
- Please explicitly define how fr is estimated from l and b.
- In equation 2 the number of transitions should be less than the number of rows. N_trans = n_rows -1, as also shown by the results (figure 15-17)
- The paper does not explicitly specify x and y are referred to body frames axis (Yr and Xr), and their definition could be misunderstood by the reader.
- Equation (4) is not clear. I would expect, seeing figure 4a, the length along each row is equal to the length of the row minus l. Please detail equation (4) derivation. In equation (4), using equation (1), b*tan(θ/2) is equal to z.
- Please detail the derivation of the translational loss area.
- Please detail how the orientation of the zig zag lines is defined as a function of the camera installation parameters. I would expect zig-zag lines parallel to the rectangle base or height. Why have the path lines a deviation angle from the rectangle base and heigth? How is this inclination angle estimated from the camera installation parameters? Please give more detailed information about path generation.
The English of the manuscript must be revised, see the following typo errors:
Line 91 – The sentence is not correct. Please check.
Line 203- remove “given” or “shown”
Author Response
"Please see the attachment."

Reviewer 2 Report
This work presents an Optimal Footprint Based Area Coverage Strategy for a False-Ceiling Inspection robot. This work is interesting and presents a mobile robotics application that could have a major impact on false-ceiling inspection. Some comments and questions that should be addressed by the authors are described below in order to improve the presentation and understanding of this work.
- One of the motivations of this work is to reduce energy consumption by using light robots without heavy batteries through optimizing the footprint area coverage. However, it would be important to comment; what is the weight of the robot with and without a high-capacity battery? Is it possible to power the robot with a cable? Since the trajectories obtained are simple and it is appreciated that a cable would not present problems. What is the maximum weight of a robot that false-ceilings could support?
- Section 4 presents multiple errors both in mathematical expressions and in the text. For example: multiple references to figures appear as "as shown in figure4 c". It is recommended to change to "as shown in Figure 4c". In the text after equation (8), it is recommended to replace "where, $ x_1, y_1, x_2, y_2, x_3 $ and $ y_3 $" by "where, $ (x_1, y_1), (x_2, y_2) $ and $ (x_3, y_3 $ ". The variables $ h $ and $ k $ do not appear in the pre-text equations. In line 213 replace "$ x, y, \ alpha, and \ betha $" by "$ x, y, \ alpha, $ and $ \ betha $".
- It is important to indicate in section 6, the resources of the PC used to carry out the analysis in Figure 11. In this same figure, the units of time are not indicated. It is assumed to be seconds.
- In section 6 the MOEA/D algorithm is sometimes also called MOEAD. They must be corrected to standardize. In similar form for NSGA2 and NSGAII.
- Captions and graphic legends do not match in Figures 12 and 13.
- Figures 6, 7, 8, 10,12, and 13 have axis labels very ambiguous, it is advisable to clarify what objective 1 and objective 2 correspond to. The descriptions in these same figures should be enriched.
- It is recommended that a more detailed study of the energy savings is obtained thanks to the optimal footprint-based area coverage strategy proposed.
- In future work, has the use of smooth trajectories been considered in order to optimize energy consumption?
Author Response
"Please see the attachment."

Round 2
Reviewer 1 Report
The authors completely ignored some of the reviewer’s comments which are reported as follows:
- Please introduce the NSGA algorithm before the results section.
- Along the paper instead of talking about optimal footprint, I would refer to camera installation parameters, which are the resulting outputs of the proposed algorithm.
In addition, the paper still lacks of detail for the path generation. By reading the text the trajectory of the robot seems to be known a priori, because the optimization algorithm only returns camera installation parameters. The authors should explicitly define in the manuscript how the path is defined. Is it predefined or it is resulting from the optimization algorithm?
Reviewer 2 Report
Most of my comments have been addressed in the first round of corrections, however, I still have some concerns.
1. Previously, ask if it is possible to power the robot with cables since the trajectories shown in figures 15b, 16b, and 17b are very simple. I assumed that being experiments, they are not complex scenarios, however, tests could be done with scenarios with more obstacles in order to highlight the usefulness of the robot and the proposed strategy.
2. It would be of great visual aid that in Figures 15, 16, and 17 a robot movement sequence is placed that coincides with the paths placed in 15b, 16b, and 17b. This is due to the fact that the images in figures 15a, 16a, and 17a do not seem to be related to any position on the path.
